# Assessing How Residual Errors of Scoring Functions Correlate to Ligand Structural Features

**DOI:** 10.3390/ijms232315018

**Published:** 2022-11-30

**Authors:** Dmitry A. Shulga, Arslan R. Shaimardanov, Nikita N. Ivanov, Vladimir A. Palyulin

**Affiliations:** Department of Chemistry, Lomonosov Moscow State University, Leninskie Gory 1/3, 119991 Moscow, Russia

**Keywords:** scoring functions, fragments, ligand structural features, errors of scoring functions, bias of scoring functions

## Abstract

Scoring functions (SFs) are ubiquitous tools for early stage drug discovery. However, their accuracy currently remains quite moderate. Despite a number of successful target-specific SFs appearing recently, up until now, no ideas on how to systematically improve the general scope of SFs have been formulated. In this work, we hypothesized that the specific features of ligands, corresponding to interactions well appreciated by medicinal chemists (e.g., hydrogen bonds, hydrophobic and aromatic interactions), might be responsible, in part, for the remaining SF errors. The latter provides direction to efforts aimed at the rational and systematic improvement of SF accuracy. In this proof-of-concept work, we took a CASF-2016 coreset of 285 ligands as a basis for comparison and calculated the values of scores for a representative panel of SFs (including AutoDock 4.2, AutoDock Vina, X-Score, NNScore2.0, ΔVina RF20, and DSX). The residual error of linear correlation of each SF value, with the experimental values of affinity and activity, was then analyzed in terms of its correlation with the presence of the fragments responsible for certain medicinal chemistry defined interactions. We showed that, despite the fact that SFs generally perform reasonably, there is room for improvement in terms of better parameterization of interactions involving certain fragments in ligands. Thus, this approach opens a potential way for the systematic improvement of SFs without their significant complication. However, the straightforward application of the proposed approach is limited by the scarcity of reliable available data for ligand–receptor complexes, which is a common problem in the field.

## 1. Introduction

Scoring functions (SFs) are ubiquitous useful tools for early stage drug discovery [1,2,3]. However, their accuracy is currently moderate and there is a clear need for an improvement in accuracy to make the entire drug discovery process less risky and demanding of experimental resources. SFs can be categorized into four distinct classes: (a) force-field or physics based; (b) empirical; (c) knowledge based (statistical); and (d) machine learning or feature based [4]. Other things being equal, the computational performance increases in a series (a)–(b)–(c)–(d), whereas the degree of generalization decreases in the same series. The classical SFs of types (a)–(c) have found and will continue to find numerous applications in drug discovery [5,6,7,8], despite all the known difficulties [1,9]. In many cases those SFs take into account (either explicitly or implicitly) the ligand–receptor affinity driving interactions, including electrostatic complementarity, hydrogen bonding and hydrophobic interactions [10]. Whereas, the traditional SFs of the first three classes seem to have reached their accuracy limits [11,12], the main recent focus is on the fourth class—the machine learning scoring functions [3]. Inspired by successes in the field of image analysis and Big Data of social media and related fields, the machine learning approaches have been given a new impetus in the fields of SF development. Higher levels of accuracy metrics have been reported for machine learning-based SFs in the literature [12,13]. 

Although machine learning approaches have definitely brought a fresh impulse in the approaches used to train SF models, the increased flexibility of those models introduced a new point of concern to the field—a greater ability of models to overfit [14,15,16,17]. Whereas this problem was less applicable to previous, rougher SF models, the machine learning approaches definitely require additional state-of-the-art efforts to ensure the resulting models are not overfitted using the available amount of input data. This is a fundamental problem in the field of drug discovery, since the amount of reliable data in comparison with the available chemical space, as pointed out by Bender et al. [18,19], is orders of magnitude less than for fields where machine learning approaches have come from and where they have had significant success. Thus, significant efforts should be applied in order to obtain a robust and not overfitted model using such flexible machine learning tools [15,20].

The field in which SF operates is intrinsically complicated—the free energy of ligand–receptor binding is affected by many different factors, their combination being different for different ligand–receptor pairs. For instance, the proper account of intramolecular ligand conformations and entropic terms was reported to be crucial [21,22]. The explicit account of water molecules is another crucial factor for certain complexes [23,24,25], but which could not be straightforwardly performed for all simulation scenarios. The intrinsic mobility of the binding site, or its parts, is another source of deviation for scoring predictions from the experimental affinity or activity, for which several approaches to sample protein conformations have been proposed [26]. The abovementioned difficulties cannot be straightforwardly solved without significant complication of the SF and, hence, decreasing its computational efficiency. The latter is the cornerstone for the main application of an SF in drug discovery practice, as an important stage in the early stages of drug discovery, where fast screening is crucial to focus the attention of researchers on a tractable fraction of large datasets of potential molecules.

Although the direct account of the complex free energy effects is cumbersome, the indirect account is quite possible, which is illustrated by the success and applicability of the target specific SFs [27]. In the latter, the parameters of the SF are specifically tuned to better reproduce the ligand–receptor interactions involving a single receptor or a limited set of receptors. Thus, the specifics of interactions, governed by the specifics of the receptor, are taken into account implicitly. The same ability of the SF to be better parameterized for a certain class of targets in comparison to the others is known to be one of the main difficulties that limits the accuracy of the “reverse screening” or ”target fishing” [8], in which a target is being predicted for a certain ligand in question. The implicit bias of the SF towards the specifics of certain types (in terms of interactions involved) of ligand–receptor complexes results in a situation whereby, for other types of ligand–receptor complexes, the prediction of the complex’s free energy appears to be systematically worse. In such a case, the choice of target for a ligand, based on the results of the virtual screening of a panel of targets, becomes complicated, since the scores that the SF produces seem to be dominated by some types of interactions. Yet another confirmation that the SF might be biased towards certain types of interactions is the better performance (in terms of robustness of predictions) of the “consensus scoring” [28], in which several different SFs have their voice in a final score value. This way, the deteriorated accuracy of one SF at certain ligand–receptor complexes is offset by the other’s SF, for which it is statistically less probable that the same type of complex is also more problematic.

In contrast to target specific SFs, the ligand-specific SFs seem to be poorly represented in the literature as everyday practical tools [29]. It can be easily explained by comparing the diversity and cardinality of the spaces of receptors and ligands. The possible diversity, and hence the accessible chemical space of ligands, is immense [30]. Thus, it is not only difficult to sample its specific subspaces adequately, but the overfit for the specifics of the ligands included into the training set is also more possible by far. The same applies to the descriptors/features defining ligand properties. The cardinality of the feature space that could reasonably explain the observed differences between different ligands of the chemical space is also large. Therefore, a large number of structural features are required to discern the properties of all ligands, even in the drug discovery related subspace. On the other hand, only a few distinct types of interactions, which are observed in the experiment and have physics-based explanations, are known and being constantly used by medicinal chemists [10]. These are, e.g., the well known hydrogen bonds, hydrophobic and aromatic interactions. Those interactions are not only well interpretable, but appear to greatly define the entire energy of the ligand–receptor interactions, which also explains their wide applicability in practice both at qualitative and quantitative levels. On the one hand, the terms of the known SFs were in many cases specifically chosen to well describe (though in a throughput manner) the abovementioned basic interactions. On the other hand, the extent to which those interactions are being properly accounted for has not been explicitly studied previously to the best of our knowledge. In a broad formulation, the question can be casted as to what extent the current SFs describe these basic interactions. At a more technical level, the question is which features of the ligands (responsible for possible interactions with the receptor) are not fully accounted for in an SF in question, and hence could be subject to a focused optimization in order to arrive at a more accurate SF. The main assumption about the possibility of improving the existing SFs is that the means of increasing the accuracy should not require additional computational overheads. Otherwise, it would limit the scope of applicability of the SFs. Thus, in the most simple and advantageous cases, only the focused parameters tuning of an SF might be required to achieve the goal.

The ability to detect the deficiencies of an SF in describing certain types of interactions represented by ligand features, therefore paves the way for the systematic studies aimed at improving the current SF and perhaps devises the ways to develop new ones with the increased accuracy. The same approach can also be used hierarchically. After the presence of the ligand features responsible for the basic types of interactions is well explained, a study of the significance of the more subtle and/or rare ligand features can be performed. For example, halogen bonding (XB) has received much attention during the last decades but is definitely not one of the main driving forces in drug discovery [31,32,33]. However, the proper account of XB by SFs might be crucial for hit-to-lead or especially lead optimization stages. Similarly, one can study various types of more specific interactions represented by certain features of the ligand. Therefore, the enhancement can be performed systematically and using the natural priorities of the significance/occurrence of the effects being taken into account.

In this work we thus hypothesized that the specific features of the ligands, corresponding to the well appreciated by medicinal chemistry interactions (e.g., hydrogen bonds, hydrophobic and aromatic interactions) might be responsible in part for the remaining SF error. The latter provides the direction for the efforts directed towards the rational and systematic improvement of the accuracy of the SFs. We also tested the proposed approach in its ability to assess the significance of the halogen bonding effect and its proper account.

In what follows, we first describe the choice of the dataset used in the study. Then, the features of the ligands, relevant for description of the basic interactions, are defined at structural level. The choice of a representative panel of the SFs is explained next. After that, a set of correlation studies is performed to reveal how the presence of the features in ligands affects the description of the experimentally measured ligand–receptor affinities. Then, the correlation of the residual errors of description of the experimental affinities (by each of the SFs in the panel) with the presence of chemical features is analyzed. Finally, several useful interpretations of the results in a broader context of drug discovery are given.

## 2. Results

To check our hypothesis on the importance of the account of specific chemical features and their contribution to the residual errors, we developed the following workflow (Figure 1). It consists of several stages, including some general QSAR procedures (such as defining, probing and filtering of features to include into the model) and building regularized regression models. At the final stage, the results provided by each of those models are interpreted in terms of the “chemical features hypothesis” mentioned in Introduction. The workflow was applied to each of the scoring functions used in this work. Detailed results are given below. A reference to the jupyter-notebook is available in Appendix A.

### 2.1. Ligand Filtering

Some of the complexes (PDB ID: 1lpg, 1oyt, 1z9g, 1ryj, 3twp, 3utu, 5c2h) turned out to be impossible to prepare using the prepare_ligand4.py program with default settings. Most of the scoring functions (i.e., AutoDock 4.2, AutoDock Vina, AutoDock VinaXB, ∆Vina RF20, and NNScore 2.0) rely on this utility as a first preparation step. Thus, for the sake of consistency, those complexes were also excluded from analysis for other SFs.

Another difficulty appeared at the stage of reading molecular structures via OpenBabel [34] Python binding library (Pybel for OpenBabel v3.1.1 [35]). It failed to correctly process ligands from 32 ligand–protein complexes (PDB ID: 1w4o, 5tmn, 1o5b, 1sqa, 1o0h, 4wiv, 2zcq, 4gr0, 1lpg, 3bv9, 4tmn, 3dxg, 1bzc, 1u1b, 4djv, 3pxf, 3utu, 1c5z, 4jia, 2zda, 3arp, 1owh, 1k1i, 3ge7, 4mme, 3ag9, 3gy4, 1o3f, 2zy1, 1vso, 2zcr, 1oyt), thus the corrections were applied via specifically developed patch procedures. First, the incorrectly perceived charges for the oxygen containing groups with the delocalized negative formal charge (O-P for phosphate and O-S for sulfate groups) were corrected. Second, the delocalized positive charge of nitrogen atoms and bond orders of N-C for amidine groups were also corrected.

Finally, only six structures (3ge7, 4djv, 4jia, 4mme, 4wiv, 3arp) with the other difficulties in reading remained, so they were also excluded from the data set. Thus, the final set included 273 complexes.

### 2.2. Statistical Analysis

Using the defined set of ligand features (see Section 4.3), the Free-Wilson (FW) type models were built using the 273 ligand–receptor complexes with well defined both experimental geometry and affinity/activity, using the coreset of CASF-2016 Update.

#### 2.2.1. Features Correlation

The mutual correlation of the features (Table 1) on a set of ligands extracted from the set of ligand–receptor complexes used was first studied (Table 2). The values of r greater than 0.5 are highlighted.

It can be seen (Table 2) that the HBD1 and HBD2 features were extremely highly correlated (r = 1). Fragments described by HBD1 appeared in more molecules (257 molecules in total) than the HBD2 fragment (256 molecules in total), and thus the HBD2 feature was excluded.

It should also be noted that Hal, F, and HP3 features were the most independent features according to r values (all of them are less than 0.50). Moreover, F and HP3 features were completely independent of each other (r = 0.00). 

In general, the mutual correlation of the proposed chemical features is not high, so we expect that the model built using these features to be statistically robust.

#### 2.2.2. Correlation of SF to the Experimental Values

Most of the selected SFs reproduced the reference *pK* values with moderate quality (*R*^2^~0.3–0.4) (Table 3) which is an expected result [11]. Most modern scoring functions are only that precise in terms of the reproduction of reference energy/*pK* values [11], which does not affect, however, the docking and ranking power of scoring function. However, it shows there is a lot of room for improvement in terms of scoring power.

Among other SFs, the ∆Vina RF20 showed somewhat outstanding performance. However, this result should be taken with care due to the partial overlap [11] of the ∆Vina RF20 training set with the currently used CASF-2016 coreset and to the known peculiarities of ML methods (greater ability to interpolate and lesser ability to extrapolate).

#### 2.2.3. Correlation of Chemical Features to the Experimental Values

It was instructive to first check our approach to see if the experimental affinity could be described by the presence of the chemical features chosen to represent the basic interactions in our study. A series of Lasso models (Table 4) with varying regularization parameters was built to check both the statistical performance of the models and which parameters are the most significant both in terms of the coefficient values and the regularization pressure they withstand (Figure 1).

Preliminary analysis shows that the most important features (according to the regression coefficients both at low and high *λ* values) were related to hydrophobic kinds of interactions (HP1, PIPI). The significance decreased in the series HP1–HP2–HP3, i.e., is inversely proportional to the bond order. Moreover, the HP3 feature, which indicates the presence of triple bonds, negatively affected binding affinity. Features representing ionic interactions (SaltBridge, PICat) are also undesirable.

Features describing halogens were shown to be important. Surprisingly, the F feature was more significant than the Hal.

Finally, hydrogen bond donors and acceptors seemingly did not play a significant role in binding, which is unexpected.

Thus, it should be expected that the scoring functions reproduce (i.e., be highly correlated to) the hydrophobic features and halogens well, and treat charged species as undesirable.

#### 2.2.4. Correlation of Chemical Features to the SF Values

It can be seen that almost all SFs (Figure 2, Table 5) except ∆Vina RF20 gave low priority to the HP1 feature, while it is of top significance according to the previously discussed results. Interestingly, the PIPI descriptor seemed to be apparently the major contributor in almost all SFs studied. It was also seen that the Hal was underrepresented in most SFs compared with its revealed significance in describing the reference values. On the contrary, the F presence was well described by most of the SFs, with the significance close to the hydrophobic terms. The latter suggests that no special treatment for fluorine interactions is necessary.

AutoDock 4.2

The weight of HP3, HP2, and PIPI interactions was comparable for AutoDock 4.2 estimations and reference *pK*. However, HP1 and Hal interactions were highly underestimated by AutoDock 4.2. It also should be noted, that AutoDock 4.2 overestimated all kinds of polar interactions, i.e., HBA, HBD1, PICat, and SaltBridge. In addition, while for the reference *pK* values PICat is considered undesirable (Figure 2), AutoDock 4.2 considers them as favorable. The same appeared for the SaltBridge.

AutoDock Vina and AutoDock VinaXB

A comparison of the regression coefficients for Vina and VinaXB showed (Table 6) that the explicit account of the halogen bonding phenomena in VinaXB did not significantly affect the quality of predictions for the current set of molecules. This may be due to the overall low number of compounds which demonstrate actual halogen bonds according to VinaXB estimations (all of them are listed in the Table 7). For other compounds, Vina and VinaXB estimations were completely numerically equal. In other words, for the selected molecule set, VinaXB predictions were generally indistinguishable from Vina predictions.

X-Score

X-Score estimated *pK* values were highly (*R*^2^ = 0.67) correlated with the Free-Wilson features. Compared with the Free-Wilson regression of the reference *pK*, X-Score underestimated the meaning of Hal and F descriptors. HP1 feature was also underestimated which is common for most of the selected scoring functions.

∆Vina RF20 and NNScore 2.0

Both ∆Vina RF20 and NNScore 2.0 are based on AutoDock Vina, but use quite different approaches to make corrections on top of it, so they pay attention to different chemical features.

∆Vina RF20 balances HP1 and PIPI weights (Figure 2, Table 5) in a ratio close to the reference (Figure 1, Table 5). It also accounts for effects caused by both heavy halogens (Hal) and fluorine (F) which also coincides with Free-Wilson coefficients for the reference.

NNScore 2.0 predictions differed significantly from other SFs. They gave meaning to insignificant features (such as HBD1, HBA) and, at the same time, did not consider important ones (Hal, HP3). Moreover, in terms of NNScore 2.0, HP3 feature was (slightly) beneficial, although it is considered not to be. The negativity of the PICat interactions was also underestimated.

DSX

DSX SF shared the general trends in correlation with ligand features as observed for many of the SFs studied. The notable exception was the HBA descriptor, which was significant for DSX values and was not so significant for the reference data description. Thus, the HBA significance seems to be overrated in DSX.

∆SAS

It is interesting that for the ∆SAS SF, contributions of the HP1 and PIPI features (Figure 2) became roughly equal, which coincides with the ratio of their contributions to the reference *pK* (Figure 1). HBA seemed to be slightly overrated compared with the correlation with the reference *pK* values.

#### 2.2.5. Correlation of Chemical Features to the Residual Error of SF Prediction

It is worth recalling that although the values of the FW matrix were normalized, the predicted *pK* values (*pK*_SF_) were not normalized, meaning that the regression coefficients of the FW descriptors and predicted *pK* should not be compared directly by their values (Figure 3). However, this also means that the closer predicted *pK*_SF_ values to the experimentally defined (reference) *pK*_ref_, the closer SF regression coefficient would be to one and the closer FW coefficients would be to zero; and vice versa, the worse the quality of the prediction made by SF, the lower would be the regression coefficient of the predicted *pK*_SF_ and the greater the corrections via FW will have to be made.

AutoDock 4.2

First, it can be seen (Figure 3) that the regression coefficient of the AD4 (0.37) was far from ideal (i.e., 1.0) and even less than 0.5.

Determination coefficient value for the combined model (*R*^2^ = 0.44) was larger than both for the single AutoDock 4.2 score (*R*^2^ = 0.31, Table 3) and FW regression (*R*^2^ = 0.36, Table 4), meaning we achieved the quality improvement by correcting the SF prediction with FW, although there is room for further improvements. This correction was mostly achieved by re-accounting of non-polar interactions (mostly HP1 and PIPI) and halogens, which is in accordance with the previously obtained results (Figure 1). The correction also tended to re-balance the polar ionic interactions contributions, which were shown (Figure 2) to be overestimated by AutoDock 4.2.

AutoDock Vina and AutoDock VinaXB

It can be seen (Figure 3, Table 8) that in combined model there was no significant difference between FW correction size for Vina and VinaXB scoring functions. This is supported by the previously demonstrated observations (Figure 2, Table 6) in which Vina and VinaXB scores were identically (qualitatively and quantitatively) reproduced by the Free-Wilson model. 

X-Score

X-Score predictions were only slightly corrected via Free-Wilson regression due to the fact that X-Score predictions themselves are strongly correlated (*R*^2^ = 0.67, Table 5) with the chemical features used in Free-Wilson regression as discussed above. A notable exception was the HP3 feature, which was not taken in account by X-Score SF. It is also interesting that HP1 correction had a small amplitude compared with the corrections for other SFs. This may be due to the well-chosen consensus hydrophobic model of the X-Score.

∆Vina RF20 and NNScore 2.0

∆Vina RF20 reached the maximal quality among the other scoring functions in the set. This is already a fairly balanced model, which does not benefit from additional account of the presence of the ligand features responsible for the intermolecular interactions using a rough Free-Wilson model.

Unlike ∆Vina RF20, NNScore 2.0 requires a significant correction coming from almost every Free-Wilson term. The use of the ML approach in NNScore 2.0 does not automatically allow it to account for specific interaction terms in a proper way. For instance, the hydrogen bond-related descriptors showed that the influence of the hydrogen bond acceptors (HBA) was overestimated, whereas the influence of the hydrogen bond donors (HBD1) was underestimated by NNScore 2.0 for proper description of the reference values. Another point of divergence is the presence of triple bonds (HP3), which require a significant negative correction.

DSX

Similar to the results of the many SFs considered, the most important correction comes from the account of hydrophobic interactions (Figure 3). The corrections Hal, F, and HP2 are next in importance. At the same time, the descriptors of polar interactions, PiCat and hydrogen bond acceptors (HBA), seem to be over-represented, necessitating a negative correction in the regression model.

It should be noted here that the amplitude and even measurement units for the DSX SF are different from the units of the reference experimental affinity of the complexes. Thus, the proposed approach, stemming from the idea of the CASF series studies [11,36] to seek a linear correlation of the SF with the reference, works consistently well with such SFs as well.

∆SAS

The results for ∆SAS SF were surprisingly similar to the results obtained for DSX. The main difference is in that the F descriptor appears to require a larger correction. Perhaps, as expected, the hydrophobic descriptors HP1 and PIPI require lesser corrections, since the change in the accessible surface upon complex formation already describes the hydrophobic interactions well.

Again, the applicability of the proposed approach is illustrated for this SF with different magnitude and units.

## 3. Discussion

The abovementioned statistical results, combined with additional reference information (Table 9), admit a reasonable interpretation and discussion, which may help to advance the field of SF development for drug discovery.

### 3.1. AutoDock 4.2

It was shown that AutoDock 4.2 SF tends to overestimate polar and ionic interactions (Figure 2) and thus requires the opposite sign correction for those components (Figure 3). This is due to the explicit treatment of electrostatic (Coulomb) interactions modeled by means of Gasteiger partial charges.

Gasteiger partial charges are known for their ability to predict and model chemical properties (such as an inductive effect). However, they are also known to be too low in amplitude (compared to any charges reasonably reproducing the electrostatic potential at HF/6-31G* level) for use in molecular mechanics applications. It was also explicitly shown [20] that the use of charge models directly reproducing the HF/6-31G* molecular electrostatic potential (MEP), in combination with robust regression analysis and outlier exclusions, improves the ability of the AutoDock 4.2 to reproduce experimental *pK* values. We assume this was not only due to the robust regression analysis of AutoDock 4.2 energy terms. Both AM1-BCC and RESP charge methods used in that work are capable of not only quantitatively reproducing the reference MEP, but also qualitatively correctly redistributing charge density compared to the Gasteiger charges, which should be especially noticeable in the case of formally charged molecules. We hypothesize that the main inconsistency in the use of Gasteiger charges for formally charged species lies in the combination of low-amplitude values of partial charge of neutral groups in combination with formally charged groups whose charge values are integers. Thus, there is no single scaling factor for these two types of groups and their respective charges. Therefore, more consistent charges between the formally charged and neutral parts of a molecule should lead to a more consistent correlation with the experimental activities.

Another point is that none of the tested scoring functions other than the AutoDock 4.2, ∆Vina RF20 and NNScore 2.0 explicitly take into account electrostatic interactions; however, they perform on the same level or even better in terms of *pK* reproduction metrics (*R*^2^, SD, Table 3). The work also showed that the most important (Figure 1) and most undervalued (Figure 3) interactions are hydrophobic in nature. Thus, the question arises: is it necessary to explicitly take into account electrostatic interactions at all? It is a known concept that the directed, in particular, electrostatic interactions are necessary not to increase affinity, but rather to ensure specificity and selectivity of binding with respect to decoy receptors. In any case, the significance of electrostatic interactions requires further detailed study.

### 3.2. AutoDock Vina and AutoDock VinaXB Halogen Bonding

AutoDock VinaXB did not show any improvement over the original AutoDock Vina. There were only 10 cases (out of 42 ligands containing heavy halogens) that exhibited non-negligible halogen bonding as assessed by AutoDock VinaXB (Table 7). However, even in these cases, the difference between the predicted *pK* values of AutoDock Vina and AutoDock VinaXB was in the range of 0.055–0.19 *pK* units, which is considered as an insignificant change (corresponding to a factor of 1.135–1.55 in K_d_/K_i_), which also does not actually lead to any increase in accuracy (Table 7).

There are two feasible hypotheses. The first is that AutoDock VinaXB is incapable of properly and fully accounting for halogen bonding. This hypothesis is partially supported by the results of Free-Wilson analysis. The second hypothesis is that it is not the halogen bonding itself that is important, but any other molecular properties of the ligand that are affected by the presence of the heavy halogen in a molecule (e.g., hydrophobicity). In any case, the topic of the importance of including of halogen bonding in scoring functions requires further research in order to narrow the gap between the general interest in XB and its proper representation in SFs.

### 3.3. X-Score

It was shown that the X-Score SF predictions themselves may be well described by Free-Wilson correlations (*R*^2^ = 0.67), which is not surprising considering that X-Score uses a linear combination of factors that account for different interactions. The latter are well described by the chemical features present in ligands. However, X-Score goes beyond (*R*^2^ = 0.41) statistics derived from a simple Free-Wilson correlation with the reference (*R*^2^ = 0.36), apparently by using a finer grained representation of the interaction, also including the receptor part. Despite its simplicity, X-Score performed as one of the best SFs in our study, which is consistent with the results of the scoring power test from the CASF-2016 Update study. It should also be noted that X-Score does not contain specific electrostatic terms other than the hydrogen bonding term and is still able to reproduce the experimental affinity well.

### 3.4. ∆Vina RF20 and NNScore 2.0

Both ∆Vina RF20 and NNScore 2.0 are machine learning SFs using the corrections based on AutoDock Vina calculations. However, they use completely different approaches to these corrections, resulting in a completely different quality of *pK* estimates.

∆Vina RF20 was shown to be superior (*R*^2^ = 0.67) among the tested SFs. Qualitatively, this is due to the correctly estimated (Figure 3) contribution of hydrophobic descriptors (especially HP1 and Hal), which were underestimated by other scoring functions in this test. Ultimately, ∆Vina RF20 does not gain any additional score from using Free-Wilson correction. This suggests that the mere presence of structural features in a ligand is not enough to improve the statistics and finer corrections are needed.

At the same time, the NNScore 2.0 estimates were rather contradictory regarding the contributions of the chemical features (Figure 2). It overestimated the features that are not important for *pK* reference reproduction (e.g., HBD1, HBA) and, at the same time, underestimated important ones (e.g., Hal, PICat, HP3). It appears that the main reason NNScore 2.0 predictions are still reasonable (*R*^2^ = 0.41) is that NNScore 2.0 is able to capture most of the hydrophobic interactions (HP1, PIPI, HP2) that have been shown to be the most important for the selected complexes set. Another possible reason is that an ensemble of models used in NNScore 2.0, even if they produce significantly different predictions, can be combined favorably in a consensus scoring model.

While ∆Vina RF20 may serve as the best example in ML class, NNScore 2.0 can serve as an example of what to expect on average. By itself, using a ML approach does not automatically increase the precision and reliability of the results. Only a wise and rigorous approach to balancing generalization and precision provides improvements. We argue that the same applies to the modification of the functional form and the parameterization of the classical SF.

### 3.5. DSX

DSX is a knowledge-based SF which does not aim at reproducing the reference energies, but instead provides a pure score. However, it can predict the experimental *pK* using linear correlation at the same quality level (*R*^2^ = 0.35, Table 3) as the scoring functions specifically designed for that purpose. Thus, the potential non-linearity of the DSX scores did not seem to show any advantages under our experiment conditions. On the other hand, the good ranking power of DSX seems to be well justified by its decent (compared with the other SFs) ability to score diverse ligand–receptor complexes.

Another, more technical point, is that the proposed approach to revealing the ligand features that are insufficiently described in SF was shown to be applicable not only to the SFs that are specifically aimed at reproducing the free energy of binding, but also to the general type of SFs that give the “score”, monotonically associated with free energy.

### 3.6. ∆SAS

∆SAS was selected as perhaps the simplest model for comparing “real” scoring functions with. It does not explicitly capture any kind of contributions other than a simple change in surface area during complex formation. However, as applied to a ligand in an already optimal position (in our case, the position extracted from crystal structures), it will characterize areas of optimal contacts and, thus, should correlate with the most important features. Indeed, the ∆SAS value was shown to be significantly better reproduced with the Free-Wilson correlation (*R*^2^ = 0.80) than for other scoring functions. The ∆SAS value, as expected, strongly correlates with the most important hydrophobic features (HP1, HP2, PIPI), so it practically does not require correction to adjust them (Figure 3). However, some polar features (PICat, HBA) and halogen features (especially F) require adjustments. 

The abovementioned findings further support that hydrophobic interactions are a major contributor to ligand–receptor affinity. Of course, as shown in the CASF-2016 Update study, this score is not sufficient to distinguish between different binding modes. This requires correct consideration of directional interactions.

∆SAS is the second SF (along with DSX) in our study, illustrating the usefulness of our approach to non-energy-based SFs.

### 3.7. The Role of Fluorine in Ligands

The fluorine atom was used as a separate feature, which became statistically significant for correlation with affinity. This reinforced, among other things, our initial assumption that the fluorine atom is commonly used in the later stages of drug design, typically to improve the ADMET properties. Despite the fact that the fluorine atom is not considered as a fragment participating in specific intermolecular interactions, the calculated value of the correlation between the presence of fluorine and experimental activity was at a good level during the study. The reason for this may be that since ADMET properties are adjusted late in the drug discovery process, the presence of a fluorine atom in the compound may indicate that the ligand is already well optimized in other directions since it has managed to reach this stage. Thus, the inclusion of fluorine atoms should not be recommended as a prospective tool to enhance affinity, as it is more of an artifact of the analyzed dataset.

### 3.8. Free-Wilson Correction

It was illustrated that Free-Wilson analysis (benchmark) of the scoring functions can be used for many purposes. First, it can be used to reveal which chemical features (i.e., interaction motives) are actually important in reproducing the reference *pK*. Second, *pK* values predicted by the scoring functions can also be decomposed in terms of the contributions of chemical features so that shortcomings in the scoring function predictions can be pre-assessed. Finally, it can be used to correct the *pK* predictions by accounting for chemical features that are underestimated by the original scoring function.

The proposed benchmark was tested in practice on several scoring functions (Table 10) and on the set of CASF-2016 complexes. The benchmark helped us to rank the chemical features in order of their actual importance (hydrophobic interactions tend to be the most important).

It was shown that the use of the Free-Wilson model, which takes into account these features on top of the scoring function, can generally improve the quality of the prediction. As a general rule, the less accurate the original model, the higher the quality can be obtained using the Free-Wilson correction (Table 9); and vice versa, the more precise and complex the initial scoring function, the less Free-Wilson approach can contribute to its quality. This is especially noticeable in the case of ∆Vina RF20. It has also been shown that some of the scoring functions may themselves correlate well to the Free-Wilson features, so their prediction will also not be improved by such a correction.

The proposed benchmark also helped to reveal inaccuracies in the accounting of these features by the selected scoring functions and, thus, outlined further directions for research and improvement.

## 4. Materials and Methods

### 4.1. Ligand–Receptor Dataset

A set of high quality ligand–receptor complex geometries with reliable binding energies data is required to study the systematic errors of the scoring functions. One of the main requirements was the availability of an experimental three-dimensional structure of the ligand–protein complex. Several databases fulfilling the requirement are known: the Protein Data Bank [44], PDBBind [45], and BindingMOAD [46].

The second important requirement is information about the experimentally measured energy, which is necessary for estimating the error of the scoring functions. This data is available in PDBBind and BindingMOAD databases. For the purpose of the work, the PDBBind database is more suitable. PDBBind includes only those complexes for which binding energy data are known. The database also contains complexes that lack missing fragments or steric overlaps. PDBBind contains a special set (PDBBind Core Set) of high quality complexes. This set was used also in the comparative assessment of scoring functions CASF-2016 Update [11], which facilitates the comparison of the results. Thus, the PDBBind Core Set was chosen as a general ligand–receptor dataset.

The PDBBind Core Set was downloaded from the PDBBind site (http://www.pdbbind.org.cn/casf.php, accessed on 13 May 2022). Structures were already prepared in this set: hydrogen atoms were present, protonation states were assigned, and all water molecules were removed from the complex structure. We used prepared molecules without additional modifications. Ultimately, [PDB_ID]_ligand.mol2 and [PDB_ID]_protein.pdb were used to estimate both SF values and construct the Free-Wilson feature matrix.

### 4.2. Scoring Function Panel

The panel of scoring functions used in the study (Table 10) was selected according to the following criteria:the availability of software implementation for academic researchers on a non-commercial basis;extensive coverage in the scientific literature and notable success stories in research and development of drug compounds;wide range of applicability with respect to ligands of various chemical compositions;It was also desirable that the final set should represent all classes of scoring functions currently identified in the literature, namely physics-based, empirical, knowledge-based, and machine learning-based.

#### 4.2.1. AutoDock 4.2

AutoDock 4.2 SF [37] implements its own force field (1). It includes vdW and hydrogen bonds energy terms; the latter is functionally similar to the former except it is also dependent on the angle. Electrostatic interactions are described by Coulomb interactions. Partial charges are calculated by the Gasteiger method [47]. In addition to the electrostatic interactions, these charges are also used to calculate the desolvation energy. Finally, the number of the rotatable bonds is also directly accounted for as a simple measure of entropy loss upon binding.
(1)$$V=w_{vdW}\overset{}{\underset{i,j}{\sum }}\left(\frac{A_{ij}}{r_{ij}^{12}}-\frac{B_{ij}}{r_{ij}^{6}}\right)+w_{H-bond}\overset{}{\underset{i,j}{\sum }}E\left(\Theta \right)\left(\frac{C_{ij}}{r_{ij}^{12}}-\frac{D_{ij}}{r_{ij}^{10}}\right)+w_{elec}\overset{}{\underset{i,j}{\sum }}\left(\frac{q_{i}\cdot q_{j}}{\varepsilon \left(r_{ij}\right)\cdot r_{ij}}\right)+w_{desolv}\overset{}{\underset{i,j}{\sum }}\left(S_{i}V_{j}+S_{j}V_{i}\right)e^{\left(-r_{ij}^{2}/2\sigma^{2}\right)}+w_{rot}N_{rot}$$

AutoDock 4.2 software implementation was accessed in Ref. [48]

Due to the artificial restrictions on the maximal number of rotatable bonds and maximal number of atoms in a macromolecule in AutoDock 4.2 source code, it had to be modified to handle molecules appearing in the CASF-2016 coreset. Thus, the source code has been modified according to the instructions in Ref. [49]. In particular, the maximum number of rotatable bonds (MAX_TORS) was increased from 32 to 40 and the maximum number of atoms (AG_MAX_ATOMS) was increased from 32,768 to 40,000. In addition, sodium parameters not present in the default AutoDock 4.2 configuration file (AD4.1_bound.dat) have been ported from another version of parameters (AD4_PARM99.dat). Additionally, the precision of the output of energy terms has been increased to 4 digits.

The calculation of AutoDock 4.2 scores requires the preparation of PDBQT protein and ligand files and the calculation of potentials grids. Preparation, including the calculation of partial charges according to Gasteiger, was carried out using the AutoDockTools (ADT) toolkit, in particular, using the prepare_ligand4.py and prepare_receptor4.py scripts with default settings. The potential grid was calculated using the autogrid4 utility. The input files for autogrid4 were prepared using the prepare_gpf4.py utility from the ADT toolkit. The grid size was chosen to be centered on the ligand (option -y in prepare_gpf4.py) and to include all its atoms with an extra space of 10Å in each (x, y, z) dimension (option -I in prepare_gpf4.py).

The input file for the autodock4 utility was generated using the prepare_dpf42.py utility (from the ADT toolkit) in single-point energy calculation mode (option -e in prepare_dpf42.py). Finally, the *pK* value was calculated using the autodock4 utility.

#### 4.2.2. X-Score

X-Score [40] (2) is one of the most widely used empirical SF.
(2)$$\Delta G_{bind}=\Delta G_{vdW}+w_{H-bond}\Delta G_{H-bond}+w_{rot}\Delta G_{rot}+w_{hydrophobic}\Delta G_{hydrophobic}$$
where Δ*G_bind_*—estimated binding energy, Δ*G_vdW_*—contribution of vdW interactions, Δ*G_H-bond_*—contribution of hydrogen bonds, Δ*G_hydrophobic_*—contribution of hydrophobic interactions, *w_i_*—regression coefficients.

X-Score exists in 3 different versions (HS, HC, HM), which implement their own methods (algorithms) to account for hydrophobic interactions. The first of these (HS) is based on the calculation of the ligand–protein contact area, which is similar to ∆SAS SF described below, except in this case only hydrophobic atoms are taken into account. The second algorithm (HC) treats hydrophobic contacts as a measure of the overlap of the vdW spheres of the hydrophobic atoms. The third method (HM) implements a hydrophobic matching algorithm that calculates the hydrophobic contribution by summing the contributions (logP) of the hydrophobic ligand atoms corresponding to the surrounding hydrophobic environment of the protein. The final binding energy is averaged over all 3 X-Score versions. This allows us to consider X-Score as a consensus model. The hydrogen bond term is the same in all three sub-methods and depends on the position and relative orientation of the atoms of potential partners in hydrogen bonds.

X-Score software implementation was accessed at Ref. [50] (v1.2).

The following command was used to calculate *pK* values using the xscore utility: “xscore -score [PDB_ID]_protein.pdb [PDB_ID]_ligand.mol2” and the value “Predicted average -log(K_d_)” was taken from the output. No special preparations of ligands and proteins were done in advance.

#### 4.2.3. AutoDock Vina

AutoDock Vina [38] is the next generation scoring function of the AutoDock family. It is highly inspired by the X-Score SF. However, some terms are different from X-Score (2). In addition to intermolecular contributions, AutoDock Vina (3) also takes into account intramolecular terms; however, the form of intramolecular terms was not described by the authors and is only available in the source code.
(3)$$\Delta G_{bind}=\Delta G_{inter}+\Delta G_{intra}\;\Delta G_{inter}=\frac{1}{1+w_{6}N_{rot}}\left[w_{1}\Delta G_{gauss1}+w_{2}\Delta G_{gauss2}+w_{3}\Delta G_{repulsion}+w_{4}\Delta G_{H-bond}+w_{5}\Delta G_{hydrophobic}\right]$$
where Δ*G_bind_*—estimated binding energy, Δ*G_gauss_*_1_*,* Δ*G_gauss_*_2_*,* Δ*G_repulsion_*—members, characterizing steric interactions, Δ*G_H-bond_*—contribution of hydrogen bonds, Δ*G_hydrophobic_*—contribution of hydrophobic interactions, *N_rot_*—number of rotatable bonds, *w_i_*—regression coefficients.

AutoDock Vina (v1.1.2) software implementation was accessed in Ref. [51]. The PDBQT input files for the ligand and protein were prepared using the AutoDockTools (ADT) toolkit [52], in particular, prepare_ligand4.py and prepare_receptor4.py scripts with default settings. To calculate *pK* values, vina utility was launched in score_only mode.

#### 4.2.4. AutoDock VinaXB (XBSF)

One of the goals of our study is a statistical assessment of importance of a more detailed account of halogen bonding (XB) in SFs. For this reason, we compare the predictions made by the very widely used AutoDock Vina SF and its XB containing counterpart, AutoDock VinaXB (XBSF) [41]. Although different SFs have been reported that explicitly account for the XB [39,53,54,55,56,57,58], the choice of XBSF is well justified in our experiment design, since only the XB part differs in the abovementioned SFs pair (4).
(4)$$\Delta G_{bind}^{\left(VinaXB\right)}=\Delta G_{bind}^{\left(Vina\right)}+\Delta G_{XB}^{}\left(distance,angle,halogen\right)$$
where $\Delta G_{bind}^{\left(VinaXB\right)}$—binding energy estimated by AutoDock VinaXB, $\Delta G_{bind}^{\left(Vina\right)}$—binding energy estimated by AutoDock Vina, $\Delta G_{XB}^{}$—XB correction, which depends on angle, distance and halogen type.

XBSF software implementation was accessed in Ref. [39]. PDBQT input files for the ligand and protein were prepared using the AutoDockTools (ADT) toolkit [52], in particular prepare_ligand4.py and prepare_receptor4.py scripts with default settings. To calculate *pK* values, vinaXB utility was launched in score_only mode.

#### 4.2.5. ∆Vina RF20

∆Vina RF20 (5) is a descriptor-based (ML) scoring function. It combines the prediction made by AutoDock Vina (empirical SF) with the prediction made by the Random Forest model considering 20 different factors (hence the RF20 in its name), including solvation and electrostatic terms similar to those used in AutoDock 4.2 (1).

∆Vina RF20 has previously been shown to be the most successful scoring function in the scoring power test in CASF-2016 benchmark. However, this result should be treated with caution due to the partial overlap [11] of the ∆Vina RF20 training set with the CASF-2016 coreset and the known peculiarities of ML methods (their greater ability to interpolate than extrapolate).
(5)$$\Delta G_{bind}^{\left(VinaRF20\right)}=\Delta G_{bind}^{\left(Vina\right)}+\Delta G_{}^{\left(RF20\right)}$$
where $\Delta G_{bind}^{\left(VinaRF20\right)}—$binding energy estimated by ∆Vina RF20, $\Delta G_{bind}^{\left(Vina\right)}$—binding energy estimated by AutoDock Vina, $\Delta G_{}^{\left(RF20\right)}$—correction made by the random forest model.

∆Vina RF20 software implementation was accessed at Ref. [59]. PDBQT input files for the ligand and protein were prepared using the AutoDockTools (ADT) toolkit [52], in particular the prepare_ligand4.py and prepare_receptor4.py scripts with default settings. For information on installation and running the ∆Vina RF20 software, see the documentation provided by the developer.

#### 4.2.6. NNScore 2.0

NNScore 2.0 is another example of descriptor-based SF in our test. It also takes AutoDock Vina prediction into account but, unlike ∆Vina RF20, it uses a different ML approach. NNScore 2.0 averages the prediction of an ensemble of 20 pre-trained neural networks that make their predictions based on AutoDock Vina term values (3) and BINANA descriptors [60]. Each of the networks was trained using its own variant of the training set.

NNScore 2.0 software implementation was accessed in Ref. [61] (v2.02). PDBQT input files for the ligand and protein were prepared using the AutoDockTools (ADT) toolkit [52], namely the prepare_ligand4.py and prepare_receptor4.py scripts with default settings. For information about installing and running NNScore 2.0 software, see the documentation provided by the developer.

#### 4.2.7. DSX (DrugScoreX)

A single Knowledge-Based SF is represented in our panel by DrugScoreX (DSX) [43] as the most available outside of commercial packages. Unlike predictions made by other functions, DSX scores are negative by default. To make comparison more even, DSX scores were taken with the opposite signs, making them positive.

DSX software implementation was accessed in Ref. [62] (v0.90).

#### 4.2.8. ∆SAS Scoring Function

A special scoring function ∆SAS estimates only the change in solvent accessible surface area (SAS) upon formation of the ligand–receptor complex. It was chosen for comparison purposes as the lower bound of quality. The idea was borrowed from the CASF-2016 Update study [11], where this SF performed perhaps surprisingly well compared to more full-featured SFs.

The ∆SAS scoring function was implemented [63] in our study using PyMOL (v2.3.0) [64] API, specifically the get_area function [65] in solvent accessible surface area (SASA) mode. The dot density parameter was set equal to 3. The radius of the solvent molecule was set equal to 1.0 Å, as in Ref. [11].

To calculate the ∆SAS value, we first calculate the SASA of the ligand molecule (ligand_sasa), the protein molecule (protein_sasa) and the entire ligand-protein complex (complex_sasa). Then the final value of ∆SAS was calculated as follows (6):(6)$$∆\mathrm{SAS}=\left(ligand\_sasa+protein\_sasa-complex\_sasa\right)/2$$

### 4.3. Fragments Related to Medicinal Chemistry Interactions

In this work, in order to search for and systematically take into account scoring function errors, it is proposed to take into account intermolecular interactions. A huge number of structurally different fragments participate in intermolecular interactions. At the same time, it is clear that not all of them can be decisive for binding. 

From the point of view of medicinal chemistry, the following interactions are usually considered: hydrogen and halogen bonds, polar, halogens and aromatic rings, hydrophobic, aryl−aryl and alkyl−aryl, cation−π [10].

The work [66] estimates the frequency of the abovementioned types of interactions in experimental ligand–protein complex geometries, including hydrophobic, hydrogen bonding, π-stacking, weak hydrogen bonding, salt bridge, amide stacking, cation–π. The most frequent found interactions are the hydrophobic interactions, followed by less frequent hydrogen bonding and π-stacking. 

Practical tools for drug discovery are also based on the same concepts of intermolecular interaction. When forming pharmacophore features, medicinal chemists use the same concepts of intermolecular interactions, including, for example, hydrogen bond donors, hydrogen bond acceptors, hydrophobic, ionic, and aromatic interactions [67,68]. Similar types of interactions, e.g., hydrogen bonds, hydrophobic and ionic interactions are the most common interactions taken into account by empirical scoring functions [4,69,70]. Aryl–aryl and aryl–alkyl interactions occur in scoring functions such as rDock [71], POLSCORE [72], ID-Score [73]. Cation–π interactions are presented in the ID-Score scoring function. Despite the main and decisive interactions seem to be well represented, the more subtle interactions require additional attention. For example, an insufficient consideration of halogen bonds in the design of new drugs is pointed out [74,75].

As a result of the theoretical and practical considerations, the following set of the basic interactions was proposed (Table 11): hydrogen bonding, hydrophobic, aryl-aryl, and salt bridge. These interactions were complemented by the finer halogen bonding and cation-π interactions, as they are also well represented in PDB complexes.

It is assumed that fluorine atoms do not form halogen bonds; therefore, fluorine atoms were not included in the main set of generalized fragments. However, the set of complexes used in the work contains ligands with fluorine atoms. As a rule, fluorine atoms are introduced into the ligand to improve the ADMET properties, in particular, to prevent metabolism at certain positions of the ligand, or to slightly increase the lipophilicity of the fragment. Fluorine, as a functional group does not carry out explicit and well interpretable intermolecular interactions with the target. Initially, the fluorine atom was not considered separately, but rather as a representative for the Hal (halogen) group. However, later we decided to isolate it because F is not known to participate in halogen bonding (XB), but it is relatively abundant in the dataset.

For all interactions considered, responsible fragments and functional groups were defined, which were then generalized and presented as a finite set using SMARTS expressions (see Table 11, #1–10). In the course of the study, a fluorine fragment was isolated as a hypothesis (see Table 11, #11, and a more detailed description above in Section 2).

### 4.4. Statistical Analysis

#### 4.4.1. Free-Wilson Analysis

To discover the dependence of the values of *pK* predicted by scoring function as well as the associated errors of prediction on the ligands chemical features, a Free-Wilson type analysis [76] was performed in the study.

First, the number of occurrences of each fragment in each molecule was counted. Correlation (7) between different features was then analyzed as a standard step for QSAR and highly correlated features (*r* > 0.9) were excluded.
(7)$$r=\frac{a\cdot b}{\left|\left|a\right|\right|\cdot \left|\left|b\right|\right|}$$
where *r*—the correlation coefficient, **a**—a vector of values of the first feature, **b**—a vector of values of the second feature.

The resulting Free-Wilson matrix contained a number of occurrences of each non-excluded feature for every ligand molecule in a set. Further, for a more meaningful comparison of the correlation coefficients for different features, the occurrence values of each chemical feature were normalized, and thus, the per-feature standard scores with zero mean and unit variance were obtained using sklearn.preprocessing.StandardScaler module of the scikit-learn (v1.1.2) library [77].

#### 4.4.2. Lasso Regression Method

The Lasso method (8) was used [78] to perform multilinear correlations between the free variables and the target value (which was either the reference or predicted *pK* value). This method was chosen both because of its controllable degree of robustness (in terms of outliers) and because of the ability to completely eliminate statistically insignificant (free) variables from the regression.
(8)$$min\left\{\frac{1}{N}{\left||y-\beta_{0}\cdot 1-X\beta |\right|}_{2}^{2}+\lambda {\left|\left|\beta \right|\right|}_{1}\right\}$$
where **y**—reference values to be predicted, X—matrix of free variables, **β**—vector of regression coefficients, *λ*—regularization parameter, **1**–unit vector, *β*_0_–the intercept.

In this work, we used a specific implementation of the Lasso method from the scikit-learn (v1.1.2) Python library (sklearn.linear_model.Lasso) [79].

#### 4.4.3. Correlation of SFs to the Experimental Values

To analyze the scoring power (i.e., the ability of the scoring function to produce binding scores in a linear correlation with experimental binding data [11]) of the selected scoring functions, we performed a linear regression and estimated its statistical properties such as determination coefficient and standard deviation.

#### 4.4.4. Correlation of the Chemical Features to the Experimental Values

To figure out which of the selected chemical features (i.e., interaction motives) determine the binding characteristics of the ligands in the selected systems, i.e., are more crucial to be properly described by a particular SF, we performed a statistical analysis of their importance by performing the Lasso regression (9) with a variable value of the regularization coefficient *λ* in the Lasso Equation (8).
(9)$$pK_{\mathrm{ref}}=Lasso\left(\mathrm{FW},\lambda \right)=\overset{N}{\underset{i}{\sum }}\beta_{i}x_{i}+\beta_{0}$$
where *pK*_ref_—reference (experimentally determined) value of *pK*, FW—Free-Wilson matrix, *λ*—regularization coefficient, *x_i_*—specific chemical feature, *N*—number of the chemical features, *β_i_*—regression coefficients.

Features which have higher regression coefficients and that are not excluded at high values of the regularization parameter *λ* are considered more statistically (and therefore chemically) important.

#### 4.4.5. Correlation of the Chemical Features to the SF Values

Then, in order to analyze which features are actually reproduced by each of the SFs, we performed a similar analysis for the predicted *pK* values (10).
(10)$$pK_{\mathrm{SF}}=Lasso\left(\mathrm{FW},\lambda \right)=\overset{N}{\underset{i}{\sum }}\beta_{i}x_{i}+\beta_{0}$$
where *pK*_SF_—value of *pK* predicted by SF, FW—Free-Wilson matrix, *λ*—regularization coefficient, *x_i_*—specific chemical feature, *N*—number of the chemical features, *β_i_*—regression coefficients.

Comparing the correlation coefficients for the *pK*_ref_ correlation with chemical features (9) and similar correlations with the SF predicted (10) *pK*_SF_ values, we can obtain a first impression of the quality of the predictions made by a particular SF.

#### 4.4.6. Correlation of the Chemical Features to the Residual Error of SF Prediction

To evaluate the deficiencies in the scoring function estimations in terms of interaction motifs (chemical features), we built a combined model that includes both the Free-Wilson matrix of the chemical features and scoring function predictions (11).
(11)$$pK_{\mathrm{ref}}=Lasso\left(\mathrm{FW}+\mathrm{SF},\lambda \right)=\overset{N}{\underset{i}{\sum }}\beta_{i}^{}x_{i}^{}+\beta_{\mathrm{SF}}pK_{\mathrm{SF}}+\beta_{0}$$
where *pK*_ref_—reference (experimentally determined) value of *pK*, FW + SF—Free-Wilson matrix supplemented with a column of the SF values, λ—the regularization coefficient, *x_i_*—a specific chemical feature, *N*—the number of the chemical features, *β_i_*—the regression coefficients for chemical features, *pK*_SF_—the value of *pK* predicted by SF, *β*_SF_—the regression coefficient for the predicted *pK*_SF_.

## 5. Conclusions

Our proof-of-concept work shows that the presence of certain features in the ligands responsible for plausible intermolecular ligand–receptor interactions does indeed correlate with the experimentally determined affinities for the CASF-2016 Update core set of ligand–receptor complexes. Moreover, in line with conventional wisdom in drug discovery, ligand–receptor affinity is dominated by hydrophobic and aromatic interactions. According to our results, the presence of charged features in ligands does not contribute to affinity. This is also consistent with both the theory where a desolvation penalty is paid and with drug discovery practice where the charged species are added to improve ADMET properties of predominantly hydrophobic molecules at some expense of their affinity [80].

The most valuable result of our study is that the residual error of the SF values relative to the experimental affinities does indeed reasonably correlate with the presence of chemical features (responsible for the basic intermolecular interactions) only in ligands from a set of ligand–receptor complexes. Moreover, different SFs show different correlation patterns of residual errors and ligand’s chemical features, thus confirming our initial assumption that SFs tend to be partially biased to better represent certain types of interaction at the expense of others. In general, we can safely state that even the basic interactions are not perfectly represented in contemporary SFs. Thus, a general approach is proposed to identify the shortcomings of SFs in terms of the description of interactions involving specific ligand’s features. This approach, combined with fine tuning tools to improve the description of problematic interactions, paves the way for the systematic study and improvement of SFs. 

However, it should be noted that the straightforward application of the proposed approach is limited by the scarcity of reliable available data for ligand–receptor complexes, which is a common problem in this area. 

## Data Availability

Not applicable.

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
