# Peer review of "Assessing How Residual Errors of Scoring Functions Correlate to Ligand Structural Features"

_ijms, 2022, doi:10.3390/ijms232315018_

Round 1

Reviewer 1 Report

In this study, the authors believe that the residual SF error may be partially explained by the unique properties of the ligands that correlate to interactions such as hydrogen bonds, hydrophobic interactions, and aromatic interactions. For proof-of-concept study, authors compare a representative panel of SFs (including AutoDock4.2, AutoDock Vina, X-Score, NNScore2.0, Vina RF20, and DSX) using the CASF-2016 coreset. According to preliminary investigation, the most significant characteristics are connected to hydrophobic types of interactions (based on the regression coefficients both at low and high values) (HP1, PIPI). The paper concludes the following important points:

The presence of charged features in ligands is not favorable for affinity according 765 to our results.

The residual error of the SF values relative to the experimental affinities does reasonably correlate to the presence of the chemical features (responsible for the basic intermolecular interactions) of only the ligands of the ligand-receptor complex set.

The paper is well written. The methods are clearly described and results are discussed in detail. Although the work is not ground breaking, the systematic analysis of scoring functions in comparison with intermolecular ligand forces is crucial and an important addition to the literature. I recommend the paper for publication after addressing couple of minor concerns.

Minor comments

The authors may add the limitation of the study in the abstract section (However it was already stated in conclusion section)

I request the authors to draw a flowchart for the overall representation of methodologies used in the present study.

In the results section, the authors discuss their observations what features are predominant in each scoring function. It is ok. In parallel, I request the authors to praise the classical scoring functions developed in the introduction section.

Author Response

Thank you for the revision. Our replies are below.

Q1: The authors may add the limitation of the study in the abstract section (However it was already stated in conclusion section)

R1: We added a short sentence on the limitation of the study to the abstract section.

Q2: I request the authors to draw a flowchart for the overall representation of
methodologies used in the present study.

R2: Thank you for the advice. We have added a flowchart to make it easier for the reader to understand the workflow.

Q3: In the results section, the authors discuss their observations what features are predominant in each scoring function. It is ok. In parallel, I request the authors to praise the classical scoring functions developed in the introduction section.

R3: We agree with the proposal and added two points into the introduction. First, we explicitly noted that the construction of the classical scoring functions has been directed to account for the most significant intermolecular interactions, though in a fast and approximate way. Second, a notion on the multiple successful applications of the classical scoring functions was also added. We believe a lot of new success stories are to come with this type of SFs despite the current focus on ML/AI-based SFs, which in turn will also bring useful results.

Reviewer 2 Report

The work on “Assessing how residual errors of scoring functions correlate to ligand structural features” by Shulga et al had established a nice correlation among Drug Structural Features vs Scoring Functions. The data set considered in the current study is limited, also mentioned by authors. The authors had some key suggestions how to improve the scoring functions in open source docking tools. The work did not consider some of the important and frequent interactions that mentioned below. It is broad interest to the medicinal chemists and can be accepted after a major revision.

Here are suggestions:

  • The authors did not consider the Anion-Pi interactions which are more frequent one.

  • It is advisable for if they compare these SF with SF in one popular commercial tool.

Author Response

Thank you for the revision. We've spent some time to think how to better proceed with the paper. Our replies are below. Generally, we hope to use your advices for the next, more in detail, paper based on broader range of ligand-receptor complexes.

Q1: The authors did not consider the Anion-Pi interactions which are more frequent one.

R1: We are aware of the anion-pi interactions and think that all interpretable interactions should be checked on potential better description by the scoring functions. However, in this study we do not mean to provide an exhaustive and production level tool, but rather the proof-of-concept approach, which, to our delight, appeared rather fruitful and general. As a result, a lot of minor interactions were intentionally left off the board since the reliable dataset of ligand-receptor complexes is scarce for decent statistics. Still we exemplify the approach by the hierarchy of the interactions by importance. So hydrophobic interactions were confirmed (in accord with the literature) to be the major driving force for ligand affinity. A few minor (less frequent and generally weaker) interactions are also considered in the study just as specific examples of applicability of the hierarchical approach (XB and Cat-pi, as well as spurious F-content). Currently, we are studying the possibility of using a more extensive dataset, which would be still reliable enough to draw statistical conclusions.
In that case, we could cover more types of different interactions known in the
literature, including Anion-pi. If we succeed, we plan to publish the findings in a more elaborate and detailed publication. We believe the current proof-of-concept work would be valuable for the scientific community.

Q2: It is advisable for if they compare these SF with SF in one popular commercial tool.

R2: Similar to the Q#1 reply, the proof-of-concept work was intended to propose an approach, not the comprehensive review. We think the findings of the work are useful and rather general, since the CASF-2016 (update) benchmark (a great work!) did not reveal any significant difference in scoring power between the commercial and open SFs. So we decided to adhere to the more easily available open SFs. Also different types of SFs were intentionally represented in order to make the results as easily comparable to the commercial SF counterparts as possible for a reader. Meanwhile, thank you for the request for the more detailed review using the proposed approach! We think that a more explicit comparison including a wider set of both open and commercial SFs could be a logical extension of the presented work. So we plan to study this option, especially, as noted above, when a more extended set of ligand-receptor complexes annotated with affinity is adopted for better statistics coverage.

Round 2

Reviewer 2 Report

The authors had provided sufficient answers to the queries that mentioned and improved the manuscript by incorporating the suggested content at the appropriate places. The manuscript can be accepted in the present form.